# In Vitro Assessment of the Effect of Implant Position on Biomechanical Behaviors of Implant-Supported Removable Partial Dentures in Kennedy Class II Condition

**DOI:** 10.3390/ma14092145

**Published:** 2021-04-23

**Authors:** Masafumi Kihara, Yoichiro Ogino, Yasuyuki Matsushita, Takehiro Morita, Yoshinori Sawae, Yasunori Ayukawa, Kiyoshi Koyano

**Affiliations:** 1Section of Implant and Rehabilitative Dentistry, Division of Oral Rehabilitation, Faculty of Dental Sciences, Kyushu University, Fukuoka 812-8582, Japan; kihara@dent.kyushu-u.ac.jp (M.K.); matsushi@dent.kyushu-u.ac.jp (Y.M.); ayukawa@dent.kyushu-u.ac.jp (Y.A.); koyano@dent.kyushu-u.ac.jp (K.K.); 2Section of Fixed Prosthodontics, Division of Oral Rehabilitation, Faculty of Dental Science, Kyushu University, Fukuoka 812-8582, Japan; 3Department of Mechanical Engineering, Faculty of Engineering, Kyushu University, Fukuoka 812-8582, Japan; morita.takehiro.871@m.kyushu-u.ac.jp (T.M.); sawae.yoshinori.134@m.kyushu-u.ac.jp (Y.S.)

**Keywords:** implant-supported removable partial denture 2, in vitro simulation model 3, Kennedy class II

## Abstract

The purpose of this study was to investigate the effects of implant position and loading position on biomechanical behaviors using implant-supported removable partial denture (ISRPD) models in a simulated Kennedy class Ⅱ partially edentulous mandible. Three types of Kennedy class Ⅱ mandibular acrylic resin models (a conventional RPD without support by an implant—CRPD; models with an implant placed at first molar (#46)—MP-ISRPD— and second molar (#47)—DP-ISRPD) were used to measure vertical displacement of the RPD, mesio-distal displacement of the abutment tooth, and bending moment of the abutment tooth and implant under one-point loading. The variables at three respective loading points (#45, #46 and #47) were compared statistically. Vertical displacement was suppressed in ISRPDs compared to the CRPD, and significant effects were identified under loading at the implant position. The largest meiso-distal displacement was observed in MP-ISRPD under #47 loading. Bending moments of the abutment tooth and implant were significantly higher in MP-ISRPD than in DP-ISPRD. In MP-ISRPD, a higher bending moment of the abutment tooth under #45 and #47 loading was detected, although the bending moment in DP-ISRPD was almost zero. The results of this study suggested that MP-ISRPD shows the specific biomechanical behaviors, although DP-ISRPD might provide biomechanical benefits under all one-point loading conditions.

## 1. Introduction

Patients with Kennedy class I or II sometimes complain about their distal extension removable partial dentures (RPDs) due to pain, poor retention and instability [1]. Subsequent edentulous ridge resorption often results in further ill fitness. Fixed implant-supported prostheses have generally been used as an alternative method for these patients. However, some patients are note able to have a sufficient number of implants because of conditions such as residual bone volume and quality. To overcome these problems, the use of implants as source of support and retention for distal extension RPD was introduced and has since been evaluated [2,3,4,5,6,7,8]. This type of prosthesis is generally called “implant-assisted RPD (IARPD)” [5,6,8]. Occlusal force acting on RPDs induces vertical displacement (tissueward movement), resulting in pain and discomfort. Support by fewer implants can prevent movement towards the tissues and these adverse phenomena. Implant-supported RPDs (ISRPDs) have been defined as RPDs with implants that provide support only [8]. ISRPDs can work with healing abutments or caps onto implants, without any specific attachments [9,10,11,12,13,14,15]. This system is very simple but has been reported to yield a stable distal extension RPD; however, some articles also have reported the positive effect of ISRPDs with attachment systems for retention, such as ball and locator attachments [11,16,17,18,19,20]. Although ISRPDs with retention—IARPDs—can be a favorable method to enhance the removal force of RPDs [21] and to improve patient satisfaction [5,10], the gradual loss of retention of these attachment systems is known as the most common complication [22,23]. In addition, the fracture of RPDs was observed in ISRPDs with attachments, although no denture fractures were found in ISRPDs without any attachments [24]. The main disadvantage of distal extension RPDs is vertical displacement; thus, ISRPDs without any specific attachments can be a simple solution from the aspect of reducing risk of complications.

Although this type of ISRPD is an obviously simple method, the designs of the prostheses, especially implant position, might have a significant effect on abutment teeth and RPD displacement. Brudvik [25] and Grossman et al. [11] noted that an implant in the second molar area was ideal in an ISRPD; however, the biomechanical evidence was lacking. Although implant position depends on the distribution of residual teeth, the evidence of optimal implant position from a biomechanical point of view is relatively insufficient.

The aim of this study was to evaluate the effects of implant position and loading position on vertical displacement of RPDs, mesio-distal displacement of the abutment tooth and bending moment of the implant and abutment tooth biomechanically, using a conventional RPD without support by an implant (CRPD) and ISRPD models in simulated Kennedy class II partially edentulous mandibles.

## 2. Materials and Methods

### 2.1. Experimental Model

A partially edentulous mandibular acrylic resin model (E50-520, NISSIN, Kyoto, Japan) was used as a simulation model. A right second premolar (#45) and two molars (#46, #47) were missing (Figure 1). The lower right first premolar (#44) root was covered with 1.0 mm thick polyvinyl siloxane impression material (Exaflex injection type, GC Corp., Tokyo, Japan) to simulate the periodontal ligament [9,15,26]. The residual ridge was also covered with 2.0 mm thick polyvinyl siloxane impression material to simulate the resilient edentulous ridge mucosa [9,15,26]. An implant (3.75 mm in diameter, 10 mm in length; Zimmer Biomet Dental, Palm Beach Gardens, FL, USA) was placed at the right first molar region, vertically to the occlusal plane (mesially positioned model; MP-ISRPD). Similarly, an implant of the same size was placed at a right second molar, vertical to the occlusal plane (distally positioned model; DP-ISRPD).

### 2.2. RPDs for Analyses

A one-piece cast removable partial denture was fabricated using dental palladium alloy (Castwell, GC Corp., Tokyo, Japan) on the simulation model. A circumferential clasp with a distal rest (an Akers clasp) was applied on a right lower first premolar as a direct retainer, and a double Akers clasp (an embrasure clasp) was applied on a lower left second premolar and a first molar as an indirect retainer (Figure 2). The denture base was made of dental acrylic resin in the usual manner.

Two types of ISRPDs (MP- and DP-ISRPD) and a CRPD were prepared for biomechanical analyses. A temporary healing abutment with a 4.0 mm height (TH254, Zimmer Biomet dental, Palm Beach Gardens, FL, USA) was applied for denture support with no retention. In accordance with the 2.0 mm thickness of the simulated mucosa, the supragingival height of the abutment for support was 2.0 mm.

### 2.3. Measuring Devices and Procedures

Stresses on the implant and tooth surface were measured using the strain gauge technique. Four strain gauges (KFR-2N-120-C1-16N10C2, KYOWA electric-corporation, Tokyo, Japan) were attached to the implant and tooth root surface (Figure 3). The electric signals from eight gauges were amplified and transmitted and then recorded [26]. A vertical load of 100 N was applied to three loading points (#45, #46 and #47) by autograph (AGS, Shimadzu, Kyoto, Japan), and the displacement of the loading device of autograph was simultaneously recorded (Figure 3) [26,27]. A concentrated load was applied with cross head speed 1 mm/min by autograph. In total, five measurements of strain and displacement at each load (each position) were recorded under the same condition. Each sequence of strain data was used to calculate the bending moment of the implant and tooth [28].

### 2.4. Statistical Analyses

The mean values and standard deviations were calculated in each analysis. Statistical analysis was performed using the one-way analysis of variance (ANOVA) with a post hoc Tukey HSD multiple comparison test. The analysis of the implant bending moment was statistically compared between MP- and DP-ISRPD using the Student *t* test; *p* < 0.05 was considered statistically significant. All statistical analyses were performed using JMP15 software (SAS Institute Inc., Cary, NC, USA).

## 3. Results

### 3.1. Vertical Displacement at Loading Point of CRPD and ISRPD

Vertical displacements of CRPD and MP- and DP-ISRPD at each loading point are shown in Figure 4. Under #47 loading, the smallest displacement was observed in DP-ISRPD. Statistical differences were observed among all groups (*p* < 0.01: DP-IDRPD vs. MP-ISRPD and CRPD, *p* < 0.05: MP-ISRPD and CRPD, Tukey HSD test). When the loading point was #46, there were statistical differences among three RPDs in the displacement (*p* < 0.01: CRPD vs. DP-ISRPD and MP-ISRPD, *p* < 0.05: DP-ISRPD and MP-ISRPD, Tukey HSD test), although no statistical differences were detected among three groups under #45 loading.

### 3.2. Mesio-Distal Displacement of Abutment Tooth (#44)

The mean values of mesio-distal displacement of the abutment tooth (#44) under loading are shown in Figure 5. The y-axis indicates the displacement of the abutment tooth (mesial: +, distal: −). Under #47 loading, the largest mesial displacement was observed in MP-ISRPD, and statistical differences were seen among all groups (*p* < 0.01: MP-ISRPD vs. DP-ISRPD and CRPD, *p* < 0.05: DP-ISRPD and CRPD, Tukey HSD test). Under #46 loading, the distal displacement of the abutment tooth was statistically greater that MP-ISRPD (*p* < 0.05, Tukey HSD test). Under #45 loading, the distal displacement of the abutment tooth was detected in CRPD, whereas the mesial displacement was observed in ISRPD groups. There were statistical differences among three groups (*p* < 0.01, Tukey HSD test).

### 3.3. Bending Moment of Abutment Tooth (#44)

The bending moment of the abutment tooth is shown in Figure 6. The mesial and distal bending moment are described as plus and minus values, respectively. In all conditions, DP-ISRPD showed the lowest bending moment in #44 abutment tooth, which was almost zero. Statistical differences were observed among the three groups in all loading conditions (*p* < 0.01, except for DP-ISRPD vs. MP-ISRPD in #46 loading, which was *p* < 0.05, Tukey HSD test).

### 3.4. Bending Moment of the Implant

The bending moment of the implant is shown in Figure 7. Bending moments in MP-ISRPD were statistically larger than DP-ISRPD under all loading conditions (*p* < 0.01, *t*-test).

## 4. Discussion

IARPD or ISRPD can overcome the shortcomings of distal extension RPDs less invasively and more economically [2,3,4,5,6,7,8]. An ISRPD in which implants provide support only, not retention, can be a simple method [9,10,11,12,13,14,15] and can avoid subsequent complications [22,23,24]. ISRPD is especially useful for the elderly, who show declined masticatory function due to denture-related problems and often cannot travel for oral care and denture adjustment. In particular, bone availability plays a major role in the rehabilitation of the posterior area, where the bone often undergoes resorption phenomena [28]. This region is often characterized by a large amount of medullary bone, and primary and secondary stability are essential to achieve implant success [29,30]. For this reason, surgical technique and implant design could help clinicians in the implant-prosthetic management of the posterior zone of the jaw [31,32]. The present study was conducted to suggest the design of ISRPDs (implant position) from a biomechanical point of view using an in vitro model.

To better understand and discuss the present results, we would like to discuss the implant position. In DP-ISRPD, vertical displacement of denture base, mesio-distal displacement of the abutment tooth, and bending moment of the implant and abutment tooth were significantly suppressed, except for vertical displacement under #46 loading and mesio-distal displacement under #45 loading. On the other hand, in MP-ISRPD, a significant effect of support by the implant was observed in vertical displacement of the denture base under #46 loading. However, under #47 loading, mesio-distal displacement and bending moment of the abutment tooth were statistically greater than in CRPD. These results suggest that occlusal loading distally from an implant might result in greater displacement of the denture base and abutment tooth and greater bending moment of the implant and abutment tooth. It follows from this suggestion that an implant should be placed as distal as possible, based on the distal occlusal force from the maxilla from the biomechanical point of view. In addition, we should notice that occlusal loading distally from an implant beneath the distal extension denture base might have an adverse impact from the biomechanical point of view. However, we would like to emphasize that these findings were observed in ISRPD without implant retention and not with an abutment for retention.

In this study, #44 abutment tooth had an artificial periodontal ligament similar to previous studies [9,15,26]. If #44 tooth had no artificial periodontal ligament, the bending moment of the abutment tooth would be higher due to its immobility, and the simulation of clinical conditions would be impossible. We understand that it would not be ideal, but it would be better for the simulation. Analyses of the bending moment of the abutment tooth showed unexpected results, especially under #45 loading (Figure 6). In the CRPD model, all loading induced vertical displacement (tissueward movement), resulting in distal bending moment in the abutment tooth. On the other hand, the displacement of ISRPD was varied by the implant beneath the denture base. In ISRPDs under #45 loading, the displacements were similar, although the distance between the implant and abutment tooth was different. We speculate that more bending of the denture base may be induced in MP-ISRPD, resulting in a greater mesial bending moment in the abutment tooth. In DP-ISRPD, the mesial displacement of the abutment tooth might be compensated by the elastic modulus of abutment tooth displacement, although further investigation is required.

The results in the present study aim to explain the concept of implant position in ISRPD. Based on the results in the present study, DP-ISRPD can support occlusal force by the implant and RPD, although this is weaker in MP-ISRPD, depending on the loading point. In addition, in MP-ISRPD, a larger displacement and bending moment of abutment teeth were observed, depending on loading points. These might be attributed to the specific biomechanical behaviors of MP-ISRPD. In DP-ISRPD, smaller adverse effects were identified due to smaller displacement.

There has been much research evaluating the effect of implant position in IARPD or ISRPD. Cunha et al. investigated the effect of implant position in ISRPD for Kennedy class II on stress distribution and RPD displacement using a two-dimensional finite element method (FEM) and concluded placement of the implant in the mesial region provided the best results when compared with those obtained in the other models; this is opposite to our results [33]. We reason that the loading was applied at five application points equally (uniformly distributed loading) in their study. In our study, one-point loading was applied to simulate the first step of mastication for crushing; this might result in a greater displacement and bending moment. Matsudate et al. reported the load distribution on the abutment tooth, implant and residual ridge in a mandibular unilateral distal-extension edentulous simulation model, which is similar to our model [26]. They reported that a higher load on the abutment tooth was observed in IARPD with an implant at the site of second molar. This suggests that IARPD with a second molar implant could enhance the occlusal force on the abutment tooth; this model is considered to support occlusal force with the implant and RPD, which might be similar to our concept. They also showed that the load on the residual ridge was lowest with DISRPD, which suggests that the displacement of RPD was the smallest in this type of IARPD. However, the effect of abutment for retention used in their study cannot be discussed here. Further studies are required to assess the effect of abutment with and without retention. Memari et al. [34] and Ortiz-Puigpelat et al. [35] also reported biomechanical behavior of IARPD using three-dimensional FEM. Their loading was uniformly distributed, and this was supposed to suppress the displacement of IARPD. Ohyama et al. used three-dimensional FEM to assess implant position and optimal implant height [36]. They suggested that higher (2 mm height) abutment could suppress the displacement of ISRPD, and the effectiveness of the distal implant was reported. In the present study, the supragingival height of the implant was almost 2 mm, which was similar to their study. The different findings between the implants placed at the first molar position were attributed to loading condition (uniformly distributed loading). Further study might show differences in various settings, especially the effects of distal loading of implants beneath ISRPD on biomechanical behavior.

The limitations associated with in vitro study, such as loading condition and mandibular acrylic resin model, must be described. Regarding the loading condition, the previous studies analyzed biomechanical behavior using one-point loading [9,12,15,21,26], uniform distributed loading [33,35] or other conditions [13,14,36]. During mastication, the distribution of occlusal force is continuously changing, and one-point loading can simulate the beginning of the mastication [13]. The dynamics of mastication are simple and stereotyped [37]. However, it is very hard to simulate a sequence of mastication. Previous studies used one of several occlusal loading conditions; this study design must be considered as a limitation. In addition, the differences between mandibular and maxillary models, including the design of dentures, were not discussed. We need to emphasize that there are differences between in vitro and in vivo models as well as simulated and actual mastication.

The other limitation is the difference between in vitro design and clinical application of ISRPD or IARPD. Previous clinical studies showed the results of IARPD with retentive attachments [16,17,18,19,20], although several studies evaluated ISPRD without retention or in combination with IARPD [5,6,10,11]. These ISRPD, not IARPD, studies revealed their positive effects. Of course, we need to know the biomechanical behaviors of IARPD (ISRPD with retention) in vitro. However, a very simple design of ISRPD might be much in demand, especially for elderly patients due to ease of management and a minimum risk of complications, as mentioned above. Some were original designs and some were used to change the design from a fixed type of prostheses. This means that further studies might be required to validate the effectiveness of ISRPD and to compare ISRPD with IARPD, including in vitro and clinical longitudinal studies from the prosthetic and hygienic points of view.

## 5. Conclusions

In our specific experimental condition, DP-ISRPD might provide biomechanical benefits for the displacement of RPD and the abutment tooth as well as the bending moment of the abutment tooth and implant. However, occlusal force during mastication is highly divergent; clinical research is required to evaluate the effect of implant position in ISRPD on biomechanical behaviors.

## Figures and Tables

**Figure 1 materials-14-02145-f001:**
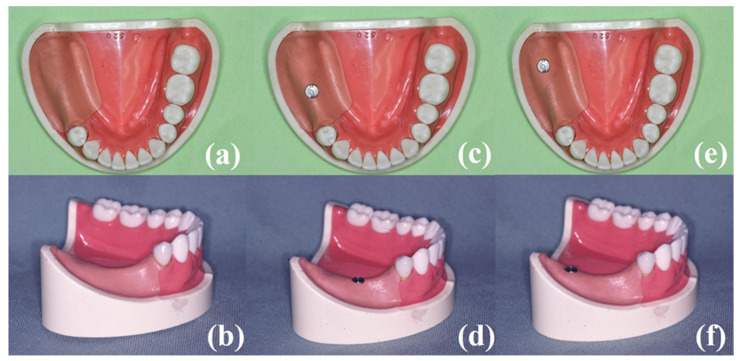
Images of simulated Kennedy class II partially edentulous mandibles: (**a**,**b**) partially edentulous mandibular Kennedy class II acrylic resin models without an implant; (**c**,**d**) mesially positioned implant-supported removable partial denture; (**e**,**f**) distally positioned implant-supported removable partial denture.

**Figure 2 materials-14-02145-f002:**
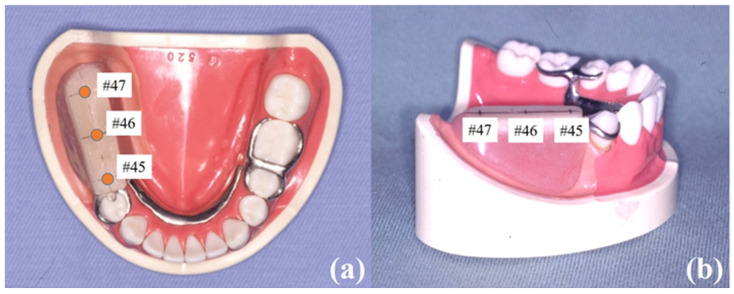
Images of simulated mandible model with a removable partial denture and loading points. (**a**) Occlusal view; (**b**) sagittal view.

**Figure 3 materials-14-02145-f003:**
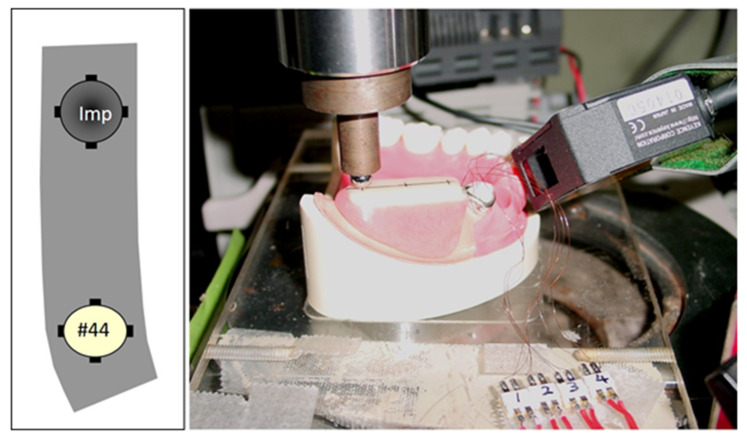
Schema and image of biomechanical analyses. Four strain gauges were attached to the implant and artificial tooth root surface. Biomechanical behaviors under a vertical load of 100 N by autograph at each loading point were recorded and calculated.

**Figure 4 materials-14-02145-f004:**
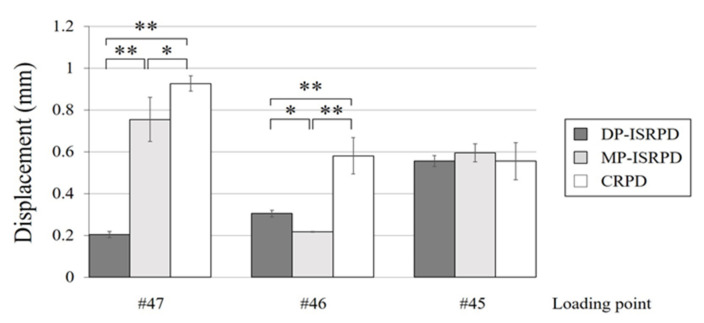
Vertical displacement (mm) of denture base under each loading point (#45, #46 and #47) in conventional removable partial denture (CRPD), mesially positioned implant-supported removable partial denture (MP-ISRPD) and distally positioned implant-supported removable partial denture (DP-ISRPD). Error bars represent the standard deviation, and statistical analyses are described as * *p* < 0.05 and ** *p* < 0.01, analysis of variance (ANOVA) with Tukey Honestly Significant Difference (HSD).

**Figure 5 materials-14-02145-f005:**
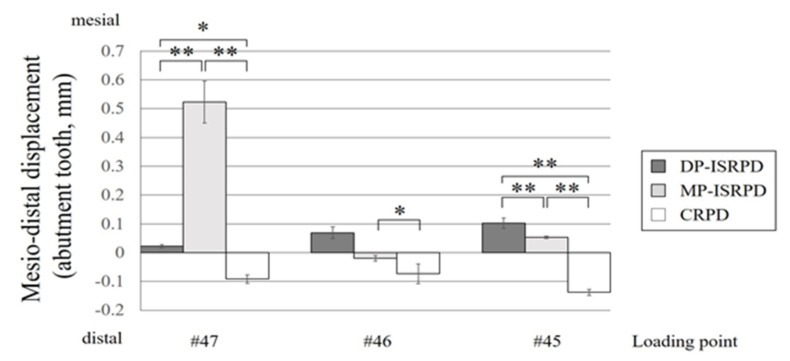
Mesio-distal displacement (mm) of abutment tooth under each loading point (#45, #46 and #47) in conventional removable partial denture (CRPD), mesially positioned implant-supported removable partial denture (MP-ISRPD) and distally positioned implant-supported removable partial denture (DP-ISRPD). Error bars represent the standard deviation, and statistical analyses are described as * *p* < 0.05 and ** *p* < 0.01, analysis of variance (ANOVA) with Tukey Honestly Significant Difference (HSD).

**Figure 6 materials-14-02145-f006:**
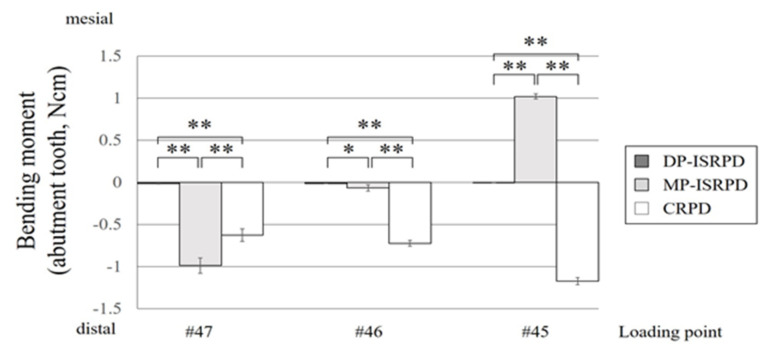
Bending moment (Ncm) of abutment tooth under each loading point (#45, #46 and #47) in conventional removable partial denture (CRPD), mesially positioned implant-supported removable partial denture (MP-ISRPD) and distally positioned implant-supported removable partial denture (DP-ISRPD). Error bars represent the standard deviation, and statistical analyses are described as * *p* < 0.05 and ** *p* < 0.01, analysis of variance (ANOVA) with Tukey Honestly Significant Difference (HSD).

**Figure 7 materials-14-02145-f007:**
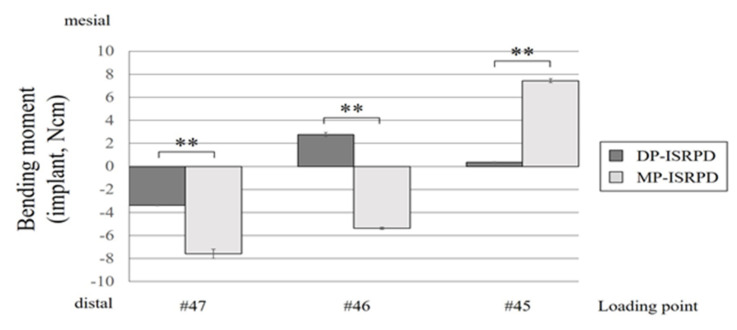
Bending moment (Ncm) of implant under each loading point (#45, #46 and #47) in mesially positioned implant-supported removable partial denture (MP-ISRPD) and distally positioned implant-supported removable partial denture (DP-ISRPD). Error bars represent the standard deviation, and statistical analyses are described as ** *p* < 0.01, *t*-test.

## Data Availability

The datasets used and/or analyzed during the current study are available from the corresponding author on reasonable request.

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
