# Peer review of "In Vitro Assessment of the Effect of Implant Position on Biomechanical Behaviors of Implant-Supported Removable Partial Dentures in Kennedy Class II Condition"

_materials, 2021, doi:10.3390/ma14092145_

Round 1

Reviewer 1 Report

The manuscript submitted to Materials entitled “In Vitro Assessment of the Effect of Implant Position on Biomechanical Behaviors of Implant-supported Removable Partial 3 Denture in Kennedy Class II Condition” is an original article which aim to investigate the design of ISRPDs (implant position) from the biomechanical point of view using in vitro model.

On my opinion the article is well written, with good English. Although it is an in vitro study on an acrylic model with little reference to clinical situations, the content of the manuscript is very interesting with clear results

The authors concluded that DP-ISRPD might provide biomechanical benefits for the displacement of RPD and abutment tooth, and bending moment of abutment tooth and implant.

However, I highlighted some issues.

In the discussion section, after “Especially, ISRPD must be useful for the elderly who show the declined masticatory 189 function due to denture-related problems and cannot often visit for oral care and denture 190 adjustment” (line189), please added these sentences with the references:
“In particular, bone availability plays a major role in the rehabilitation of the posterior area, where the bone often undergoes resorption phenomena [PMID: 15185985]. This region is often characterized by a large amount of medullary bone and primary and secondary stability are essential to achieve implant success [PMID: 19025637 - PMID: 32098046]. For this reason, surgical technique and implant design could help clinicians in the implant-prosthetic management of the posterior zone of the jaws [PMID: 16231543 - PMID: 32475099].”

There is a lack of references to clinical scenarios.

Line 220 – “The results in the present study clearly indicate the concept..” Please modify “clearly indicate” with “aim to explain the concept …”

Line 228 – “In this section, the differences between the present study and some of 228 the previous studies are discussed” Please, delete this sentence. It is unnecessary.

At the end of discussion section add the limitation of this study:

- The one-point loading vs uniform distributed loading

- The use of mandibular acrylic resin model, different from clinical scenarios

- The absence of difference between maxilla and mandibular models

- The absence of in vivo results

Please, highlight that in the clinical scenarios occlusal force during mastication is uniformly distributed.

Author Response

Reviewer 1

The manuscript submitted to Materials entitled “In Vitro Assessment of the Effect of Implant Position on Biomechanical Behaviors of Implant-supported Removable Partial 3 Denture in Kennedy Class II Condition” is an original article which aim to investigate the design of ISRPDs (implant position) from the biomechanical point of view using in vitro model.

On my opinion the article is well written, with good English. Although it is an in vitro study on an acrylic model with little reference to clinical situations, the content of the manuscript is very interesting with clear results

The authors concluded that DP-ISRPD might provide biomechanical benefits for the displacement of RPD and abutment tooth, and bending moment of abutment tooth and implant.

However, I highlighted some issues. 

Point 1: In the discussion section, after “Especially, ISRPD must be useful for the elderly who show the declined masticatory 189 function due to denture-related problems and cannot often visit for oral care and denture 190 adjustment” (line189), please added these sentences with the references:
“In particular, bone availability plays a major role in the rehabilitation of the posterior area, where the bone often undergoes resorption phenomena [PMID: 15185985]. This region is often characterized by a large amount of medullary bone and primary and secondary stability are essential to achieve implant success [PMID: 19025637 - PMID: 32098046]. For this reason, surgical technique and implant design could help clinicians in the implant-prosthetic management of the posterior zone of the jaws [PMID: 16231543 - PMID: 32475099].”

> Thank you for your meaningful comments. In this part, we revised the description as you suggested. It must be helpful to understand the background of this study  

There is a lack of references to clinical scenarios.

Response 1: As you suggested, the comments regarding clinical scenarios were insufficient. As other reviewers also mentioned, the clinical studies to assess the effectiveness of ISRPD without retention were a few, although many in vitro studies used this type of model. This ISRPD without retention must be very simple and might be expected to use more for elderly patients. In addition, the simulation of actual mastication is very difficult and we added some comments in Discussion section as one of clinical scenarios. Please recheck the body text.

Pont 2: Line 220 – “The results in the present study clearly indicate the concept..” Please modify “clearly indicate” with “aim to explain the concept …”  Line 228 – “In this section, the differences between the present study and some of 228 the previous studies are discussed” Please, delete this sentence. It is unnecessary.

Response 2: As you can see the text, we follow your advices.

Point 3: At the end of discussion section add the limitation of this study:

- The one-point loading vs uniform distributed loading

- The use of mandibular acrylic resin model, different from clinical scenarios

- The absence of difference between maxilla and mandibular models

- The absence of in vivo results

Response 3: As you suggested we added some comments regarding the limitations of this in vitro study.

Point 4: Please, highlight that in the clinical scenarios occlusal force during mastication is uniformly distributed.

Response 4: As we mentioned above, we added some comments about the simulation of mastication. In addition, we added some comments about the future demand of this type of ISRPD which might be expected in the future, especially for elderly. Unfortunately, a limited number of clinical study has been reported, although a number of in vitro studies have used this type of ISRPD. I hope you can understand these background.

Reviewer 2 Report

The question of this in vitro work has only a low clinical relevance. Treatments of this type are rare. 
In the discussion, please give a much larger part to the classification of your work in clinical studies.

You yourself state that these types of restorations are made with retentive abutments (locator or ball abutment). 
It would be interesting to compare the measurements as you have performed them with retentive elements. 
The simulation of the periodontal ligament with the covering of the root with an elastic material is a massive simplification of the mechanical properties of the periodontium. Please discuss this and critically evaluate the results on tooth 44 accordingly.

Author Response

Reviewer2

Point 1: The question of this in vitro work has only a low clinical relevance. Treatments of this type are rare. 
In the discussion, please give a much larger part to the classification of your work in clinical studies.

Response 1: Thank you for your comments. As you suggested, the number of the previous clinical studies that used this type of ISRPD was small, although many in vitro studies used this design. However, the demand or clinical application of this simple ISRPD might be expected, especially for elderly patients. These comments were added in Discussion. I hope you can understand these background.

Point 2: You yourself state that these types of restorations are made with retentive abutments (locator or ball abutment). 
It would be interesting to compare the measurements as you have performed them with retentive elements. 

Response 2: We completely agree your comments. This was the first study to show the results of the simplest in vitro study. As you comment above, ISRPD with retentive attachments or IARPD have been evaluated in the previous clinical studies. We understand the studies focused on IARPD must be required. This must be next purpose to be elucidated. We mentioned in Discussion.

Point 3: The simulation of the periodontal ligament with the covering of the root with an elastic material is a massive simplification of the mechanical properties of the periodontium. Please discuss this and critically evaluate the results on tooth 44 accordingly.

Response 3:  The in vitro simulation of the periodontal ligament has been reported so far as you can see the references. We added some comments at the beginning of 3rd paragraph in Discussion.

Round 2

Reviewer 1 Report

The authors evidenced the limits of this study.
The content of the manuscript was uploaded following the reviewers suggestions enriching the scientific soundness.
Clinical relevance is still low, however the manuscript can give rise to reflections on the various rehabilitation techniques.

Reviewer 2 Report

The concerns from the review were put into perspective by the authors' responses.
The clinical relevance remains low, of course, but cannot be solved in any other way.